# Highly Efficient Blood Protein Analysis Using Membrane Purification Technique and Super-Hydrophobic SERS Platform for Precise Screening and Staging of Nasopharyngeal Carcinoma

**DOI:** 10.3390/nano12152724

**Published:** 2022-08-08

**Authors:** Jinyong Lin, Youliang Weng, Xueliang Lin, Sufang Qiu, Zufang Huang, Changbin Pan, Ying Li, Kien Voon Kong, Xianzeng Zhang, Shangyuan Feng

**Affiliations:** 1Key Laboratory of OptoElectronic Science and Technology for Medicine, Ministry of Education, Fujian Provincial Key Laboratory for Photonics Technology, Fujian Normal University, Fuzhou 350007, China; 2Clinical Oncology School of Fujian Medical University, Fujian Cancer Hospital, Fuzhou 350014, China; 3Fujian Provincial Key Laboratory for Advanced Micro-Nano Photonics Technology and Devices, Research Center for Photonics Technology, Quanzhou Normal University, Quanzhou 362046, China; 4Department of Chemistry, National Taiwan University, Taipei 10617, Taiwan

**Keywords:** protein SERS, super-hydrophobic platform, deep learning, nasopharyngeal carcinoma

## Abstract

Early screening and precise staging are crucial for reducing mortality in patients with nasopharyngeal carcinoma (NPC). This study aimed to assess the performance of blood protein surface-enhanced Raman scattering (SERS) spectroscopy, combined with deep learning, for the precise detection of NPC. A highly efficient protein SERS analysis, based on a membrane purification technique and super-hydrophobic platform, was developed and applied to blood samples from 1164 subjects, including 225 healthy volunteers, 120 stage I, 249 stage II, 291 stage III, and 279 stage IV NPC patients. The proteins were rapidly purified from only 10 µL of blood plasma using the membrane purification technique. Then, the super-hydrophobic platform was prepared to pre-concentrate tiny amounts of proteins by forming a uniform deposition to provide repeatable SERS spectra. A total of 1164 high-quality protein SERS spectra were rapidly collected using a self-developed macro-Raman system. A convolutional neural network-based deep-learning algorithm was used to classify the spectra. An accuracy of 100% was achieved for distinguishing between the healthy and NPC groups, and accuracies of 96%, 96%, 100%, and 100% were found for the differential classification among the four NPC stages. This study demonstrated the great promise of SERS- and deep-learning-based blood protein testing for rapid, non-invasive, and precise screening and staging of NPC.

## 1. Introduction

Nasopharyngeal carcinoma (NPC) is a highly aggressive malignancy that is widespread in Southeast Asia and southern China [1], with approximately 133,000 new cases and 80,000 associated deaths annually [2]. Clinical staging (I, II, III, and IV), based on the tumor–node–metastasis (TNM) system, is crucial for guiding clinicians in their treatment decisions for different risk groups, and it is the most important survival predictor for NPC patients [3]. According to the National Comprehensive Cancer Network guidelines (2021 version), the recommended treatment for stage I NPC is radiotherapy alone; for stage II, it is concurrent chemoradiotherapy; while for stage III–IV NPC, it is induction chemotherapy with concurrent chemoradiotherapy [4]. Correspondingly, the five-year overall survival rate for stages I and II NPC can reach up to 98% and 92%, respectively. However, this will drop to 83% and 71% for stages III and IV NPC, respectively, as previously reported by our group [5]. Furthermore, the more intense radiation and chemotherapy for advanced (stages III and IV) NPC may lead to serious treatment-related complications such as deafness, dysphagia, or temporal-lobe necrosis. This compromises the quality of life of patients, and it may even cause their death. Therefore, early screening and accurate staging play important roles in increasing therapeutic efficacy and reducing complications and mortality among NPC patients.

The conventional methods for NPC detection are nasopharyngoscope examination, histopathological examination of a biopsy specimen, the blood Epstein–Barr virus (EBV) test, and imaging examinations based on computed tomography, magnetic resonance, or positron emission tomography. Nevertheless, these methods have certain deficiencies [6,7]. For example, the nasopharyngoscope combined with histopathological examination is the gold standard in diagnosing NPC, but it is painful, invasive, time-consuming, and has a lower sensitivity for early-stage NPC, when tumors are usually very small and insidious, meaning that patients are often largely asymptomatic. The EBV test suffers from a high misdiagnosis rate and large test variations between hospitals. Imaging examinations are expensive, and they are not suitable for point-of-care tests and mass screening of all suspected cases. As a result, 70% of NPC patients present with advanced disease at first diagnosis [8], and this results in a poor prognosis and serious treatment-related complications. It would thus be invaluable to develop an efficient, cheap, sensitive, and non-invasive method for the early diagnosis and precise staging of NPC in high-risk areas such as southern China.

Surface-enhanced Raman scattering (SERS) spectroscopy, which is an optical analysis method based on inelastic scattering and localized surface plasmon resonance, has been widely used in blood-based liquid biopsies for bioanalysis and medical diagnosis due to its high sensitivity, rapidness, simplicity, and specificity for identifying the structural features of biomolecules, such as proteins and nucleic acids [9,10,11,12]. In particular, label-free SERS, which can directly detect bioanalytes after adsorption onto a nanostructured gold or silver surface, is a convenient and cost-effective approach that, in contrast to label-based SERS strategies, does not require complicated sample preparation or expensive Raman tags [13,14]. Recently, label-free blood SERS analysis has blossomed into an area of intensive research for cancer diagnosis, and it has been shown to be of significant clinical value for nasopharyngeal [15], gastric [16], thyroid [17], prostate [18], lung [19], and breast cancers [20]. A label-free blood SERS method for NPC detection has also been applied by our group to analyze and identify the different tumor (T) stages. However, the diagnostic accuracy for distinguishing stage-T1 from stage-T2–T4 NPC is only 63%, and this is not satisfactory [6]. 

Blood is a complex system containing various electrolytes, saccharides, and uric acid, and a direct blood SERS analysis can easily be disrupted by these substances, resulting in limitations in the signals from plasma proteins that involve many tumor biomarkers, and are closely linked to the tumorigenesis and the progression of NPC [8,21]. Herein, to further improve the accuracy of the label-free blood SERS method for the early diagnosis and staging of NPC, a cellulose acetate (CA) membrane-based label-free SERS strategy is proposed for the simple and rapid analysis of plasma proteins of NPC patients at different TNM stages.

Considering that the concentrations of plasma proteins purified by a CA membrane are extremely low (only about 1.67–2.00 g/L), which will cause poor signal repeatability and an incomplete reflection of cancer information, we prepared a super-hydrophobic platform to pre-concentrate the proteins by forming a uniform deposition instead of the non-uniform coffee-ring [22] distribution for acquiring repeatable protein SERS spectra. Furthermore, since traditional confocal micro-Raman-system-based SERS detection (with micron-level resolution) requires an optimization of the measurement location—which is subjective and time-consuming—we developed a macro-Raman signal-acquisition system with a millimeter-level (about 3 mm) excitation laser focus and an automatic mobile platform to achieve large-scale, intelligent, and rapid SERS detection of plasma proteins in normal samples and those from the four stages of NPC. Finally, a deep-learning algorithm based on a convolutional neural network (CNN) was employed for the precise screening and staging of NPC.

To the best of our knowledge, this is the first report on label-free plasma protein SERS detection combined with a deep-learning algorithm for the purposes of early detection and clinical staging of NPC. Such detection and accurate staging are crucial for clinicians to precisely assess a patient’s status at an early stage and make optimal treatment decisions with the lowest incidence of treatment complications and the best clinical effects for different risk groups.

## 2. Materials and Methods

### 2.1. Collection of Human Blood Plasma

This research included 225 healthy volunteers and 939 NPC patients, confirmed by a histopathological diagnosis at Fujian Cancer Hospital (Fuzhou, China) between September 2016 and October 2020. All subject groups underwent a nasopharyngoscopy. The study was guided by the Declaration of Helsinki and was approved by the ethical committee of Fujian Cancer Hospital (No. SQ2022-073-01). Informed consent was obtained from all subjects. At the first diagnosis, whole blood was collected with the use of EDTA anticoagulant, and this was centrifuged at 2000 rpm for 15 min at 4 °C to obtain blood plasma. The plasma was stored at −80 °C before use. According to the *American Joint Committee on Cancer Staging Manual (8th edition)* [23], 120 patients in the study were stage I, 249 were stage II, 291 were stage III, and 279 were stage IV. Detailed patient information is shown in Table 1.

### 2.2. Preparation of Super-Hydrophobic Platform and Ag NPs

An aluminium (Al)-sheet-based super-hydrophobic platform was prepared using a simple, fast, low-cost approach based on chemical etching and chemical vapor deposition (Figure 1A). Unlike the prior scheme [24], in which a costly 0.8 cm-thick aluminium plate along with a complicated time-consuming mechanical polishing step are required, we employed a cheap 0.05 cm-thin aluminium sheet as the substrate and simplified the preparation process by omitting the polishing step while adjusting the parameters in the chemical etching step in this study. In detail, first, an Al sheet (17.0 cm × 17.0 cm × 0.05 cm) was pressed into 100 matrix-curved grooves (1.5 mm in depth and 8 mm in diameter) using a pressing machine for the sake of sample self-localization and high-throughput detection. Second, each groove in the Al sheet was cleaned with ultrapure water, acetone, and ethanol, in turn, in an ultrasonic bath for 5 min and then dried at room temperature. Third, 100 µL HCl solution (5 mol/L) was added to each groove for chemical etching for 1 h. Fourth, the groove was modified by trimethoxy (1H,1H,2H,2H-heptadecafluorodecyl) silane with a chemical vapor deposition in enclosed conditions at 120 °C for 2 h to obtain a super-hydrophobic surface (Figure 1B). A scanning electron microscope (SEM) micrograph of the prepared super-hydrophobic platform showed an irregular lamellar micro-nanometer morphology (Figure 1C), which can increase the contact area with the air and promote the concentration process [25]. In this study, the water contact angle of the surface was 151° (Figure 1D), indicating good super-hydrophobicity.

Colloidal silver nanoparticles (Ag NPs) were prepared by the reduction of silver nitrate with hydroxylamine hydrochloride, as reported by Leopold et al. [26]. The colloidal solution was a milky gray color. The Ag NPs were spherical shapes and were 35 ± 5 nm in diameter with an absorption peak at 418 nm, which were characterized by transmission electron microscopy (TEM) and an ultraviolet–visible (UV) absorption spectrum (Figure 1E,F). Moreover, the zeta potential was −21.32 mV (Figure 1G). Before use, the colloidal silver was centrifuged (10,000 rpm for 8 min) to remove the supernatant.

### 2.3. Purification of Plasma Protein

The plasma protein was rapidly purified (within 15 min) using a membrane purification technique, as shown in Figure 2A. Specifically, 10 μL of blood plasma was first dropped onto the CA membrane for adsorption. After this, the CA membrane was rinsed in a washing solution (glacial acetic acid, 95% ethanol, and distilled water in the ratio 1:9:10, respectively) to eliminate the non-protein components in the plasma. Then, the membrane containing the plasma protein was cut and collected into centrifuge tubes. Subsequently, with continual stirring, acetic acid (300 mL) and the Ag NP solution (300 mL) were added to dissolve the CA membrane and enhance the protein SERS signal, respectively. Finally, the protein–Ag NP mixture in the supernatant solution was deposited onto the super-hydrophobic platform using a pipette tip and air-dried to pre-concentrate the protein for the SERS measurements.

### 2.4. Construction of High-Throughput, Rapid Macro-Raman System

A high-throughput, rapid macro-Raman system was constructed for the protein SERS measurements, as shown in Figure 2B. In brief, the system comprises: a constant diode laser (785 nm, 100 mW); a near-infrared (NIR) spectrograph (Acton LS-785, Princeton Instruments); a back-illuminated, NIR-optimized, deep-depletion CCD camera (1340 × 400 pixels, pixel size: 20 μm × 20 μm, Model PIXIS 400BR, Princeton Instruments, Trenton, NJ, USA); a self-designed fiber optic Raman probe for both the signal excitation and collection; and an automatic mobile platform, coupled with automatic spectrum-acquisition software, for rapid and intelligent sample detection.

The Raman probe consists of six surrounding collection fibers (300 µm in diameter) and a central excitation fiber (200 µm in diameter). The ends of the collection and excitation fibers are coated with a long-pass (LP) filter and a short-pass (SP) filter, respectively. The SP filter in the excitation path only allows 785 ± 2.5 nm laser light to pass through, limiting any interference from the stray light inside the fiber. The LP filter in the collection path, which has a cut-off wavelength of 805 nm, only allows the Raman signal to be transmitted to the spectrometer, preventing interference from light reflected from the sample. For this custom system, the focused laser spot is about 3 mm in diameter. This can cover the whole pre-concentrated sample (1.5 mm in diameter) and provide Raman signals with an 8 cm^−1^ spectral resolution in the range of 400–1800 cm^−1^ in only a 1 s integration time.

### 2.5. Construction of Deep-Learning Model

As described, a total of 1164 protein SERS spectra were acquired from the participants listed in Table 1. The original SERS spectra were preprocessed by fluorescence background subtraction and area normalization. Firstly, the fluorescence background was subtracted by the fifth-order improved modified multi-polynomial fitting algorithm developed by Zhao [27]. This algorithm took into account the effects of the noise level and peak contribution, thereby suppressing the undesirable artificial peaks that may occur in polynomial fittings. After fitting, the area normalization method was implemented to normalize the integrated area under each SERS spectrum (from 400 to 1800 cm^−1^) to a value of 1. In this way, the “absolute intensity” in the SERS spectrum was replaced by “relative intensity”. Thus, the absolute intensity variations from laser fluctuations and sample concentration inhomogeneity can be eliminated, which helps in comparing the variation in relative compositions and structures among different samples. After this, the preprocessed spectra were randomly split into two groups: a training dataset (about 80% of the spectra) and an independent test dataset (about 20% of the spectra) for statistical analyses. The training dataset (180 healthy volunteers, 96 stage I NPC, 201 stage II NPC, 234 stage III NPC, and 225 stage IV NPC) was used to establish a prediction model using a deep-learning algorithm for NPC screening and staging; the test dataset (45 healthy volunteers, 24 stage I NPC, 48 stage II NPC, 57 stage III NPC, and 54 stage IV NPC) was used to assess the performance of the prediction model, as shown in Figure 2C.

The deep-learning model used in this study was constructed using a one-dimensional five-layer CNN. As shown in Figure 2D, the CNN contained, in turn, an input layer, two stacked convolutional layers (16 kernels, size: 3 × 1) activated with a hyperbolic tangent (tanh) function, a max-pooling layer (size: 3 × 1, stride: 1), two stacked convolutional layers (32 kernels, size: 3 × 1) activated with a tanh function, a max-pooling layer (size: 3 × 1, stride: 1), two stacked convolutional layers (64 kernels, size: 3 × 1) activated with a tanh function, a max-pooling layer (size: 3 × 1, stride: 1), two stacked convolutional layers (128 kernels, size: 3 × 1) activated with a tanh function, a max-pooling layer (size: 3 × 1, stride: 1), two stacked convolutional layers (256 kernels, size: 3 × 1) activated with a tanh function, a max-pooling layer (size: 3 × 1, stride: 1), a flatten layer, a fully connected layer, and a softmax output layer. The model was optimized with an Adam optimizer and a cross-entropy loss function. There were 100 training epochs, the batch size was 20, and the learning rate was 0.001. The CNN algorithm was implemented using the Keras library and the TensorFlow framework.

## 3. Results and Discussion

Figure 3A shows a Ag-NP-based SERS spectrum from plasma protein purified by a CA membrane (red line), a SERS signal from a blank CA membrane with Ag NPs (blue line), and the background Raman signal from the Ag NPs (black line). These were collected under the same experimental conditions. Comparing these three spectra, we can see that there are many intense and sharp molecular vibration bands in the protein SERS spectra, while there are hardly any Raman peaks from the blank CA membrane or the Ag NP substrate. This indicates that the use of the CA membrane and Ag NPs in the experimental process does not introduce background interference in the range of interest, from 400 to 1800 cm^−1^.

To examine the repeatability of the protein SERS spectra on the super-hydrophobic platform, 10-μL samples of the protein–Ag NP mixture were dropped onto both the super-hydrophobic Al platform and an ordinary Al plate, and air-dried. The dried area of the mixture on the super-hydrophobic Al platform (about 1.5 mm in diameter) was smaller than that on the ordinary Al plate (about 5.5 mm in diameter, insets of Figure 3B,C), revealing that the protein was concentrated on the super-hydrophobic Al platform. Spectra were then obtained from five random points along the diameters of the dried areas of the mixtures on the two substrates (Figure 3B,C). It can be seen that the SERS spectral intensities and shapes from the sample on the ordinary Al surface vary considerably, which indicates poor signal repeatability and an incomplete reflection of cancer information. This phenomenon was attributed to the very low concentration of plasma proteins from the CA-membrane purification and the unevenly dispersed distribution of the protein–Ag NP mixture on the ordinary Al surface. In contrast, the reproducibility of the SERS signal on the super-hydrophobic Al surface was improved enormously due to the pre-concentration of proteins and the uniform deposition. The reproducibility was also evaluated by recording the randomly collected 30 SERS spectra from one plasma protein sample. The relative standard deviation (RSD) value of only 1.39% was obtained for the intensity of the characteristic peaks at 1003 cm^−1^, showing wonderful repeatability. Furthermore, the detection sensitivity can reach 10^−12^ M by measuring the SERS spectra of R6G with different concentrations, from 10^−10^ M to 10^−12^ M, on the super-hydrophobic substrate. This improves the reliability and efficiency of detection, meaning that this type of SERS analysis could be applied to mass screening and the clinical staging of NPC.

Based on the self-developed high-throughput macro-Raman system and super-hydrophobic platform, stable and complete protein SERS signals for each subject were rapidly acquired with an integration time of only 1 s. Figure 3D shows the normalized average protein SERS spectra from the healthy volunteers and patients at each of the NPC stages, along with their corresponding difference spectra. The average protein SERS spectra of both the healthy and NPC subjects displayed numerous characteristic peaks, including at 490, 548, 625, 645, 679, 740, 756, 836, 851, 881, 936, 1003, 1033, 1046, 1124, 1180, 1209, 1266, 1313, 1449, 1523, 1553, 1616, and 1679 cm^−1^, with the strongest signals at 1003, 1266, 1449, and 1679 cm^−1^. To better understand the molecular basis of these protein SERS spectra, the tentative assignments are listed in Table 2 [28,29]. The spectra display most of the information from various amino acids. The difference spectra, in Figure 3D, reveal changes in the spectral intensities between the healthy and NPC blood proteins. Figure 3E,F show the SERS spectral differences among the average protein SERS spectra from the different clinical stages of NPC.

To present the differences in these characteristic peaks more clearly, statistical comparative analyses between two or more groups were conducted using the Mann–Whitney U or Kruskal–Wallis tests as appropriate. Figure 4 shows a box plot comparison of the SERS spectral intensities between the normal and NPC groups. When compared with normal samples, we found that the NPC blood protein samples exhibited higher intensities at 548, 756, 1003, 1033, 1046, 1124, 1266, 1553, and 1679 cm^−1^, while they were lower at 490, 625, 645, 740, 836, 851, 881, 936, 1180, 1209, 1313, 1449, and 1523 cm^−1^. Moreover, as shown in Figure 5 and Appendix A, the intensities of these SERS peaks also exhibited statistically significant changes with the cancer staging developments from stage I to stage IV; some, such as 1003 and 1033 cm^−1^, were observed to change linearly.

These spectral differences reflect the alterations in blood protein composition and conformation with the progress of nasopharyngeal neoplasia. For instance, the SERS bands of arginine (490, 851, and 881 cm^−1^) and tyrosine (1180 and 1209 cm^−1^) in the NPC blood protein showed lower intensities than those in the normal controls, indicating a decrease in the abundance of certain amino acids relative to the total SERS-active components in NPC subjects. In previous work, our group also found a decrease in these two amino acids in the blood of patients with colorectal cancer [30]. In addition, the higher spectral intensities of tryptophan (at 548, 756, 1046, 1266, and 1553 cm^−1^) and phenylalanine (at 1003 and 1033 cm^−1^) for NPC subjects suggest that there is an increase in the tryptophan and phenylalanine contents relative to the total protein components in the NPC blood, which is in agreement with our previous preliminary SERS analyses of blood and saliva samples from NPC patients [15,31]. Furthermore, increased Raman signals for tryptophan and phenylalanine have also been identified in other tumor tissue and blood [32,33]. Tryptophan plays an important role in T-cell proliferation, and phenylalanine is among the eight essential amino acids. The higher levels of tryptophan and phenylalanine may therefore be related to the immune rejection reactions that can cause the elimination of cancer cells [33]. 

This study is the first to show that the Raman peaks of phenylalanine at 1003 and 1033 cm^−1^ increase with the progress of NPC from stage I to stage IV, implying that phenylalanine may be a potential biomarker for the clinical staging of NPC at different risk levels. In a serum nuclear magnetic resonance-based metabolic study, Jobard et al. also reported that an increased phenylalanine content is associated with advanced breast cancer in comparison to early breast cancer [34]. 

In this study, NPC was also accompanied by decreased SERS bands related to glutathione (at 625 and 645 cm^−1^). Glutathione is an antioxidant that plays an important role in cellular detoxication processes for quenching endogenous peroxides and free radicals [35]. The decreased glutathione content may therefore be associated with a reduction in the body’s ability to counter increased oxidative stress in the tumor microenvironment, which may activate the process of tumorigenesis. A decreased glutathione content has also been observed in previous research examining other malignancies, such as laryngeal, cervical, and breast cancer [36,37]. The content of glutamate (740, 1313, and 1523 cm^−1^) and glycine (936 cm^−1^), which are the components of glutathione, were also found to decrease in our study.

The bands for proline (836 and 851 cm^−1^), alanine (851 and 1449 cm^−1^), valine (1124 cm^−1^), amide III (1266 cm^−1^), and amide I (1679 cm^−1^) were significantly down- or up-regulated in the cancer group, indicating that there are alterations in the specific amino acid contents and the secondary structure of plasma protein in tumorigenesis and the progression of NPC. The reasons for this may be as follows: NPC is an Epstein–Barr virus (EBV)-associated tumor, and EBV-related antibodies (such as EBV EA-IgA and VCA-IgA) are overexpressed in NPC blood. In addition, other blood proteins, such as p16 (cyclin-dependent kinase inhibitory protein), p53 (nuclear protein), and Hsp90α (heat-shock protein 90α), have also been found to be significantly down- or up-regulated in the tumorigenesis and progression of NPC, resulting in alterations of the composition and conformation of total blood protein, and in turn, accelerating the progression to advanced cancer [21,38]. Our group has also found changes in protein conformation in colorectal and liver cancer patients [39,40]. These variations in protein SERS spectra among the different TNM stages of NPC patients and healthy subjects indicate the great potential of CA-membrane-based label-free protein SERS for NPC screening and detection at different TNM stages.

It is worth noting, that the above simplistic peak-intensity analysis involves only finite information, and there were significant variations and overlapping intensities in the protein SERS spectra of the four cancer stages and the healthy control group. It is thus necessary to employ powerful diagnostic algorithms to make the best use of the complete spectral signals and extract more information for the precise screening and staging of NPC.

CNN-based deep-learning algorithms are advanced artificial-intelligence systems with good performance in recognizing images or spectral data for disease detection [41,42]. For a Raman spectral analysis, CNN algorithms have been used to successfully distinguish liver cancer [41], prostate cancer [43], and breast cancer [44] from control groups. However, these studies only focused on cancer screening and did not consider clinical staging (multi-classification detection), which is closely correlated with the survival rates of cancer patients. In this study, with the development of the high-throughput macro-Raman detection system, we were able to rapidly acquire protein SERS spectra from patients at different clinical stages of NPC. These were then analyzed by a CNN algorithm to identify the clinical staging of NPC. This is extremely valuable for efficiently assessing the progress of NPC and making optimal treatment decisions for different risk groups before treatment.

For the CNN statistical analysis in our study, as described, the SERS spectra were split into standard 80% and 20% training and test sets, respectively. The SERS spectra from the healthy volunteers and NPC patients at stages I–IV were labeled 0, 1, 2, 3, and 4, respectively, for classification. As shown in Figure 6A, the CNN model converged quickly during training; the prediction loss dropped sharply, and the accuracy increased dramatically over a relatively short period of the training epochs. The cross-entropy loss was computed to reflect the error between the predicted and true labels of the spectra. The trend of the accuracy curve was contrary to the loss curve because a lower loss indicates a better fit to the set. After the 55th epoch, the loss was close to the plateau, near to zero, and the accuracy reached 100% on the training set, revealing a good prediction performance. This finding is in good agreement with a fivefold cross-validation accuracy on the training dataset, which was 100% (Figure 6B).

In the test stage, double-blind tests were carried out with the 228 reserved test spectra (45 healthy, 24 stage I, 48 stage II, 57 stage III, and 54 stage IV NPC). After placing these test spectra into the CNN model, the accuracies for the healthy patients and stages I–IV in the NPC groups were 100%, 96%, 96%, 100%, and 100%, respectively, also indicating relatively satisfactory classification results. These statistical results are summarized in Figure 6C.

To further evaluate the efficacy of the CNN model, we compared it with a traditional algorithm, namely the principal component analysis and linear discriminant analysis (PCA-LDA). As shown in Figure 6D–F, for the training set, PCA-LDA obtained training accuracies of 100%, 85.4%, 85.6%, 88.0%, and 92.4% for the healthy volunteers and NPC stages I–IV, respectively. For the test dataset, the testing accuracies of PCA-LDA for the five groups were 100%, 70.8%, 72.9%, 77.2%, and 87.0%, respectively. We found that both the CNN and the PCA-LDA algorithms were able to precisely identify NPC with 100% accuracy, showing that the protein SERS spectra acquired by the membrane purification technique and super-hydrophobic SERS analytical platform had excellent performance for NPC screening. However, for NPC staging, PCA-LDA performed poorer than the CNN model, with considerable overlaps among the four clinical stages of NPC (as shown in Figure 6D), especially for stage I NPC detection (only 70.8% accuracy on the test set). CNN-based deep-learning algorithms can be used to classify massive and complex datasets, addressing the complexity and heterogeneity limitations [45] and making them more accurate for predicting multi-type samples than the PCA-LDA algorithm. Consequently, our constructed CNN model showed a better performance algorithm for NPC staging, illustrating the great potential of the label-free blood protein SERS-based CNN model for the precise screening and staging of NPC.

## 4. Conclusions

In summary, we developed a highly efficient blood protein SERS analysis method based on a membrane purification technique, a super-hydrophobic platform, and a rapid macro-Raman system to precisely screen and classify NPC at different clinical stages, which is crucial for reducing complications and mortality. The super-hydrophobic platform can pre-concentrate tiny amounts of blood proteins by forming a uniform deposition, and this provided repeatable SERS spectra. The spectral results indicated alterations in the blood protein composition (e.g., amino acids content) and conformation during the tumorigenesis and progression of NPC. A CNN-based deep-learning algorithm was successfully employed to classify protein SERS spectra of healthy volunteers and NPC subjects at the four different stages, with good classification performance (up to 100%). Overall, this SERS- and deep-learning-based blood protein test shows promising potential for a rapid, non-invasive, and precise screening and staging of NPC.

## Figures and Tables

**Figure 1 nanomaterials-12-02724-f001:**
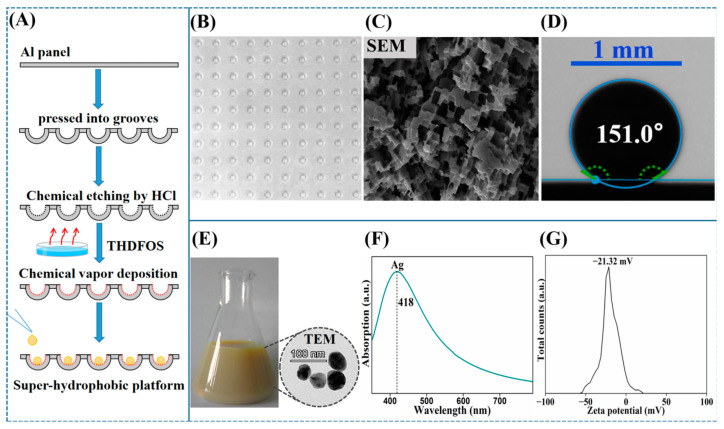
(**A**) Schematic of the procedure for preparing the super-hydrophobic platform by chemical etching and chemical vapor deposition. (**B**) Photograph of Al sheet-based super-hydrophobic platform. (**C**) SEM micrograph of the super-hydrophobic grooved surface. (**D**) Optical image of a water droplet on the surface. (**E**) Photograph and TEM micrograph of the Ag NPs. (**F**) Ultraviolet–visible (UV) absorption spectrum of the Ag NPs. (**G**) Zeta potential of the Ag NPs.

**Figure 2 nanomaterials-12-02724-f002:**
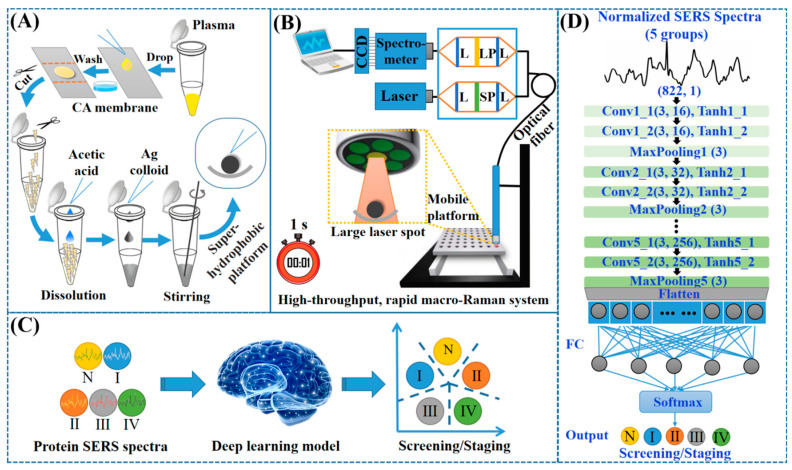
(**A**) Schematic of CA-membrane-based purification of plasma protein for SERS detection. (**B**) High-throughput, rapid macro-Raman system (abbreviations: L, lens; SP, short pass; LP, long pass). (**C**) Protein SERS spectra for NPC screening and staging using a deep-learning algorithm, and (**D**) structure of CNN-based deep-learning model (abbreviations: Conv, convolutional layer; FC, fully connected layer; N, normal; I, stage I NPC; II, stage II NPC; III, stage III NPC; IV, stage IV NPC).

**Figure 3 nanomaterials-12-02724-f003:**
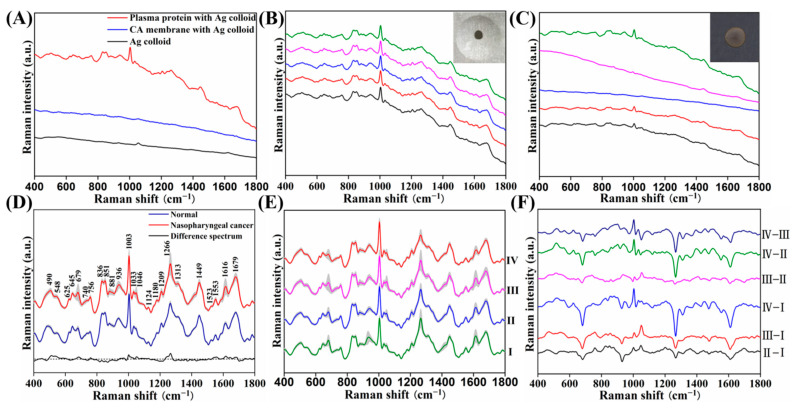
(**A**) Comparison of a SERS spectrum from plasma protein purified by CA membrane, a SERS signal from a blank CA membrane, and the background signal from the Ag NPs. (**B**,**C**) Spectra from five random sampling points along the diameters of the dried mixtures on the super-hydrophobic and the ordinary Al surfaces, respectively. (**D**) Comparison of the normalized average protein SERS spectra from healthy and all NPC subjects, and the corresponding difference spectrum (NPC minus normal). (**E**) Average protein SERS spectra from NPC subjects from the four stages, and (**F**) their corresponding difference spectra.

**Figure 4 nanomaterials-12-02724-f004:**
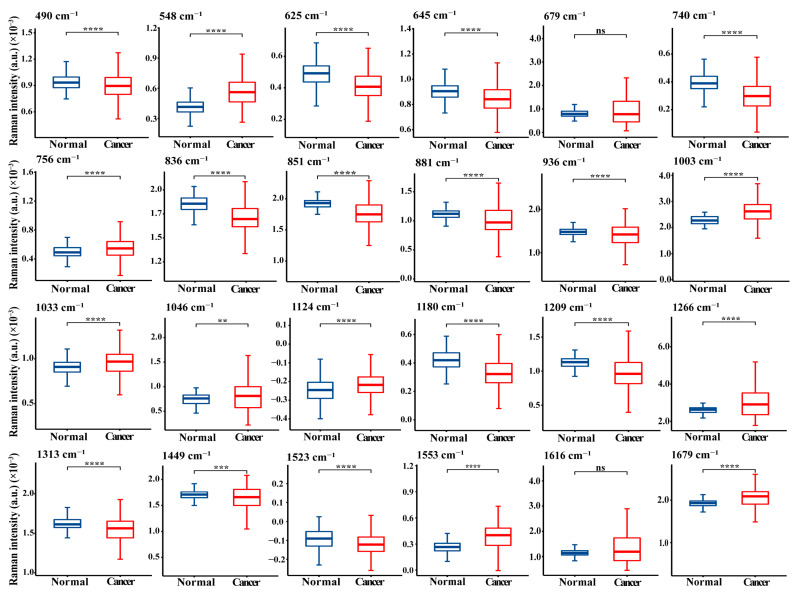
Box plots of the peak intensities of protein SERS spectra for the normal volunteers and NPC subjects at all stages. For each box, the central line represents the median, and the lower and upper boundaries indicate the 25th and 75th percentiles, respectively. Abbreviations: ns indicates no significance; ** *p* < 0.01; *** *p* < 0.001; **** *p* < 0.0001 (Mann–Whitney U test).

**Figure 5 nanomaterials-12-02724-f005:**
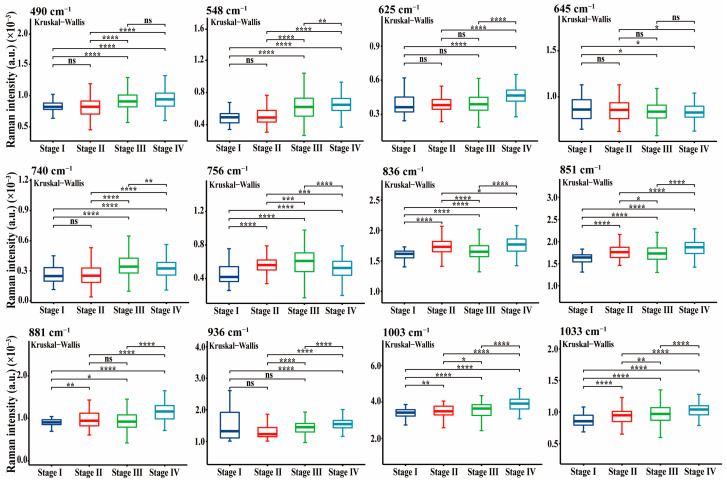
Box plots of some of the protein SERS peak intensities for NPC subjects at each of the four stages. Abbreviations: ns indicates no significance; * *p* < 0.05; ** *p* < 0.01; *** *p* < 0.001; **** *p* < 0.0001 (Kruskal–Wallis test).

**Figure 6 nanomaterials-12-02724-f006:**
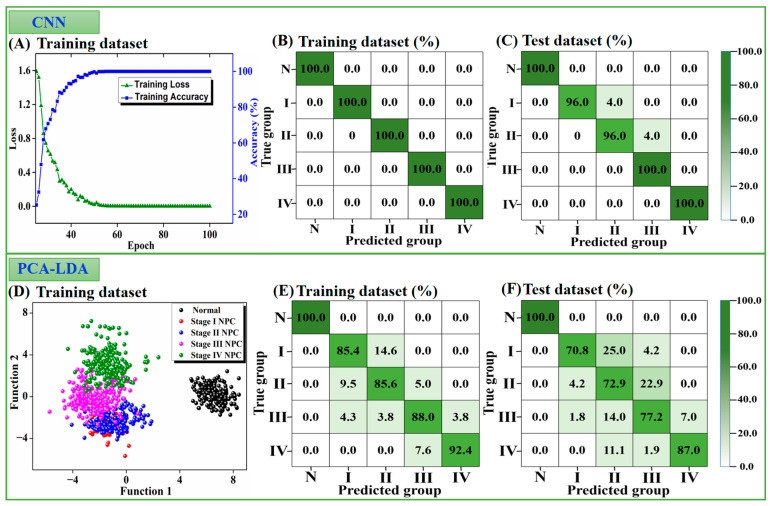
(**A**) Loss and accuracy curves of the CNN model on the training set. (**B**) Training accuracy and (**C**) testing accuracy of the CNN method for healthy and stage I, II, III, and IV NPC groups on the training and test set, respectively. (**D**) Scatter plot of the discrimination scores by PCA-LDA on the training set. (**E**) Training accuracy and (**F**) testing accuracy of PCA-LDA for the five groups.

**Table 1 nanomaterials-12-02724-t001:** Clinical information on NPC and healthy subjects.

	NPC Subjects (*n* = 939)	Healthy Subjects (*n* = 225)
Age		
Mean	45 ± 8	41 ± 11
Gender		
Male	490	123
Female	449	102
TNM stage		
I	120	N/A
II	249	N/A
III	291	N/A
IV	279	N/A

Note: N/A indicates not applicable.

**Table 2 nanomaterials-12-02724-t002:** Tentative protein assignments for the SERS spectra.

Raman Shift (cm^−1^)	Tentative Assignment	Raman Shift (cm^−1^)	Tentative Assignment
490	Arginine	1033	Phenylalanine
548	Tryptophan	1046	Tryptophan
625	Glutathione	1124	Valine
645	Glutathione	1180	Tyrosine
679	Histidine	1209	Tyrosine
740	Glutamate	1266	Amide III, Tryptophan
756	Tryptophan	1313	Glutamate
836	Proline	1449	Alanine
851	Alanine, Proline, Arginine	1523	Glutamate
881	Arginine	1553	Tryptophan
936	Glycine	1616	Serine
1003	Phenylalanine	1679	Amide I

## Data Availability

The data presented in this study are available upon request from the corresponding author.

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
