# Peer review of "Highly Efficient Blood Protein Analysis Using Membrane Purification Technique and Super-Hydrophobic SERS Platform for Precise Screening and Staging of Nasopharyngeal Carcinoma"

_nanomaterials, 2022, doi:10.3390/nano12152724_

Round 1
Reviewer 1 Report
Paper: [nanomaterials-1793384] Highly Efficient Blood Protein Analysis using Membrane Purification Technique and Super-Hydrophobic SERS Platform for Precise Screening and Staging of Nasopharyngeal Carcinoma
The authors described a SERS-based method for blood protein detection. The method first involves the purification of blood protein on a cellulose acetate membrane through adsorption and washing. Next the membrane was cut and dissolved in solution with the addition of AgNPs and dropped onto an aluminium panel-based super-hydrophobic for SERS signal acquisition. A deep learning algorithm was used to classify the raw Raman spectra for nasopharyngeal carcinoma diagnosis and staging with high accuracies. Overall, the manuscript is well written with comprehensive data. There are some comments to be considered and addressed in the manuscript for improvement. More detailed comments are as follows:
Major Comments:
1. The authors have recently reported the use of the same super hydrophobic SERS substrate for disease detection (Lin etal, 2021, Adv Funct. Mater.). This paper should be cited and the differences/improvements in this work should be discussed in this manuscript.
2. Could the authors provide evidence that non-protein molecules have been removed from the CA membrane after washing and discuss how that could affect the resultant SERS measurements?
3. The detection sensitivity and reproducibility of this method should be provided and discussed against existing label-free SERS assays for blood samples in literature.
Minor Comments:
1. Given the large laser spot in the macro-Raman system, would sample degradation be an issue to affect signal reproducibility?
Reviewer 2 Report
The result of this manuscript is quite fascinating. As I am not familiar with the medical aspect, I can only ask technical questions related to SERS.
(1) It is quite challenging to obtain SERS spectra of target molecules, proteins, in a complex media such as blood. There are any number of matters likely to interfere with adsorption of proteins onto Ag nanoparticles. Is use of a CA membrane really sufficient to remove interference?
(2) Use of a superhydrophobic surface is certainly a convenient way to pre-concentrate samples. One problem, however, occurs when one tries to transfer an aqueous sample onto such a surface, short of using an injector as with an ink jet nozzle. Did you transfer samples using a pipette tip?
(3) Even use of a superhydrophobic surface does not guarantee uniform drying. How did you manage it?
(4) Spreading a 785 nm excitation laser, 100 mW, over a spot size of 3 mm does not seem to provide enough power, even with SERS, to allow you to obtain high quality spectra with 1 second. Did you use a particularly high sensitivity detector?
(5) Presence of NaCl in samples can greatly affect the SERS peak intensity. Have you looked into this effect?
Reviewer 3 Report
Dear Editor,
The paper from Lin et al is a well described strategy to use SERS and machine learning to discern patients with cancer from healthy people. The work is well written, in a good English and with clear objectives and method described. I think the authors should better describe how they treated the SERS spectra, because it is only generally stated at lines 192-193. They should try to be more precise on how they treated the data because I think this is a fundamental step for the following analysis by machine learning. At the same time, I suggest this paper to be transferred from this journal to a special issue on Biosensors, MDPI (Surface Enhanced Raman Biosensors), because the entire paper is mostly focused on the development of a point of care and on the analysis and not on the nanomaterials. Indeed, all the characterizations of the new samples are missing. Only a TEM is reported, but it is not enough. So, I think this paper cannot be published in Nanomaterials without the characterizations of the material (major revisions), but it would be ok for the publication in the above-mentioned special issue.
Round 2
Reviewer 1 Report
The authors have addressed the review comments well and the manuscript is now recommended for publication.
Reviewer 2 Report
Thank you for providing additional information. I think this will be beneficial to readers.
Reviewer 3 Report
Dear authors, thanks for your revised version of the paper. At now I think it is ready to be published in Nanomaterials.